# A Screening of Epigenetic Therapeutic Targets for Non-Small Cell Lung Cancer Reveals PADI4 and KDM6B as Promising Candidates

**DOI:** 10.3390/ijms231911911

**Published:** 2022-10-07

**Authors:** Jéssika Cristina Chagas Lesbon, Taismara Kustro Garnica, Pedro Luiz Porfírio Xavier, Arina Lázaro Rochetti, Rui Manuel Reis, Susanne Müller, Heidge Fukumasu

**Affiliations:** 1Laboratory of Comparative and Translational Oncology, Department of Veterinary Medicine, Faculty of Animal Science and Food Engineering, University of São Paulo, Duque de Caxias, 225-Jardim Elite, Pirassununga 13635-900, SP, Brazil; 2Molecular Oncology Research Center, Hospital de Amor, Antenor Duarte Viléla, 1331-Dr. Paulo Prata, Barretos 14784-400, SP, Brazil; 3Life and Health Sciences Research Institute (ICVS), School of Health Sciences, University of Minho, 4710-057 Braga, Portugal; 4ICVS/3B’s—PT Government Associate Laboratory, 4710-057 Braga, Portugal; 5ICVS/3B’s—PT Government Associate Laboratory, 4805-017 Guimarães, Portugal; 6Structural Genomics Consortium, Buchmann Institute for Molecular Life Sciences, Johann Wolfgang Goethe University, Max-von-Laue-Str 15-60438, 60438 Frankfurt am Main, Germany

**Keywords:** NSCLC, epigenetic targets, metastasis, KDM6B, PADI4

## Abstract

Despite advances in diagnostic and therapeutic approaches for lung cancer, new therapies targeting metastasis by the specific regulation of cancer genes are needed. In this study, we screened a small library of epigenetic inhibitors in non-small-cell lung cancer (NSCLC) cell lines and evaluated 38 epigenetic targets for their potential role in metastatic NSCLC. The potential candidates were ranked by a streamlined approach using in silico and in vitro experiments based on publicly available databases and evaluated by real-time qPCR target gene expression, cell viability and invasion assays, and transcriptomic analysis. The survival rate of patients with lung adenocarcinoma is inversely correlated with the gene expression of eight epigenetic targets, and a systematic review of the literature confirmed that four of them have already been identified as targets for the treatment of NSCLC. Using nontoxic doses of the remaining inhibitors, KDM6B and PADI4 were identified as potential targets affecting the invasion and migration of metastatic lung cancer cell lines. Transcriptomic analysis of *KDM6B* and *PADI4* treated cells showed altered expression of important genes related to the metastatic process. In conclusion, we showed that KDM6B and PADI4 are promising targets for inhibiting the metastasis of lung adenocarcinoma cancer cells.

## 1. Introduction

Lung cancer is the second most diagnosed cancer worldwide and was responsible for 1,796,144 deaths in 2020, according to GLOBOCAN [1]. Non-small cell lung cancer (NSCLC) represents 85% of the cases, of which 80% are adenocarcinomas (AdCs), adenosquamous carcinomas, or squamous-cell carcinomas (SqCCs) [2,3]. Unfortunately, almost 50% of lung cancer cases are metastatic resulting in a poor prognosis and limited therapeutic options, with a critical five-year overall survival (OS) of only 10% and 1% in patients with stage IVA and IVB respectively [3,4,5]. The low survival rates are mainly attributed to chemoresistance, low detection rate of mutations in target genes, compromised choice of targeted therapy, and late diagnosis of lung cancer patients [3,6,7]. Therefore, the identification of novel therapeutic targets and their inhibitors is urgent [8]. 

Proteins that modify the epigenetic code are promising targets for the development of new anti-metastasis and anti-invasion drugs for NSCLC [8]. Histone posttranslational modifications (PTMs) represent epigenetic modifications that are frequently altered in cancer and contribute to tumor migration, metastasis, and aberrant cellular growth [9]. Many histone deacetylase (HDAC) inhibitors (HDIs), such as vorinostat and panobinostat, have shown promising results in preclinical and clinical investigations of NSCLC [9] and new molecules for epigenetic targets are being developed and explored for their use in the treatment of diverse cancers [10]. However, there remains a need to validate these targets in large-scale clinical trials [8].

The Structural Genomics Consortium (SGC) is an international public–private partnership with the goal of supporting research for a better understanding of human disease biology and to enable the discovery of new medicines (https://www.thesgc.org, accessed on 25 January 2019). To this end, SGC develops and makes available highly specific inhibitors (chemical probes) to the scientific community [11,12,13]. Most available epigenetic probes are inhibitors of bromodomains (BRDs) and protein methyltransferases (PMTs). These molecules have been shown to be effective in several tumor models by inhibiting or attenuating several characteristics relevant to tumor development, such as metastatic capacity and resistance to conventional treatment [14,15,16]. However, epigenetic inhibitors used at adequate doses can inhibit tumorigenic phenotypes without being overtly cytotoxic to cells from healthy tissues [17]. 

Here, we performed a streamlined set of in silico and in vitro experiments to rank and validate epigenetic targets that regulate the metastatic process in lung cancer cells, using SGC chemical probes as specific inhibitors.

## 2. Results

### 2.1. Evaluation of Epigenetic Targets in Non-Small Cell Lung Cancers Using Publicly Available Data

All 38 epigenetic targets were analyzed and ranked based on the significance of inverse association between survival of patients with NSCLC and gene expression. According to the selection criteria, from 1082 patients, 590 and 492 patients were selected for adenocarcinoma and squamous cell carcinoma, respectively. 

Of the 38 epigenetic targets, eight were inversely associated with low survival of lung adenocarcinoma patients (Hazard Ratio (HR) > 1, *p* < 0.05, Table 1), and none were associated with patients with squamous cell carcinoma (*n* = 492).

### 2.2. Systematic Literature Review Analysis of Potential Epigenetic Targets

The systematic review initially resulted in 98 publications related to the epigenetic target Enhancer Of Zeste 2 Polycomb Repressive Complex 2 Subunit (EZH2), 6 related to Bromodomain Containing 4 (BRD4), 5 related to Protein Arginine Methyltransferase 1 (PRMT1), 4 related to each Lysine Demethylase 6B (KDM6B) and Bromodomain Containing 9 (BRD9) targets; one related to Coactivator Associated Arginine Methyltransferase 1 (CARM1), and no work related to Bromodomain Adjacent To Zinc Finger Domain 2 (BAZ2A) or Peptidyl Arginine Deiminase 4 (PADI4) as targets in lung cancer. Specific targets, such as PRMT1, KDM6B, CARM1, BRD4, and EZH2, have been shown to be associated with malignant phenotypes of lung cancer, influencing cell proliferation and metastatic processes (regulation of epithelial–mesenchymal transition and cell invasion). Lysine Demethylase 6A (KDM6A) expression was not correlated with poor survival in lung cancer patients.

Thus, we selected the targets that presented the highest risk rate (HZ) and the lowest number of publications in the literature (<5), with the aim of studying potential new epigenetic targets for lung cancer. Therefore, the epigenetic targets selected for further analysis were KDM6B, CARM1, BAZ2A and PADI4.

### 2.3. In Silico Analysis of Expression of the 4 Target Genes in Lung Cancer Cells

Through the analysis of gene expression in silico, three cell lines showed elevated expression of the chosen potential targets (A549, H2126 and H1568). The cell lines (H2126 and H1568) were collected from metastatic sites, pleural effusion, and lymph nodes, whereas A549 cells were collected from the primary tumors (Appendix A). However, the H2126 cell line does not have the potential for in vitro invasion [18]. The expression of the epigenetic targets in healthy lung tissue showed the following results: KDM6B (Z-score = 2.8), CARM1 (Z-score = −0.5), BAZ2A (Z-score = −1.0) and PADI4 (Z-score = 0.58), Z-score < 5, suggesting non-expression in healthy lung tissue.

### 2.4. Epigenetic Targets Gene Expression by Real-Time PCR

The expression of *CARM1*, *BAZ2A*, *KDM6B* and *PADI4* was evaluated in the cell lines H2126, H1568 and A549. All three cell lines, A549, H2126, and H1568, showed a higher level of target expression in general (Appendix A).

### 2.5. Cytotoxic Potential (IC_50_) and Determination of the Maximum Concentration without Cytotoxic Effect of Epigenetic Molecules

Cytotoxic potential of the four epigenetic inhibitors, TP-064 (CARM1), GSK2801 (BAZ2A/B), GSK-J4 (KDM6A/B), and GSK484 (PADI4) in A549 cells was assessed GSK-J4 (KDM6A/B) had an IC_50_ value of 8.21 µM, whereas the other probes were not cytotoxic, even at 10 µM. Therefore, the inhibitors showed low cytotoxic potential, even when using very high doses not recommended for use, demonstrating high safety. However, we selected a concentration of 1000nM for further experiments (Figure 1).

### 2.6. Epigenetic Inhibitor Effects on Cancer Cell Migration and Invasiveness

Inhibition of the epigenetic targets KDM6A/B and PADI4 reduced cell invasiveness compared to the control group (*p* < 0.05) in the A549 and H1568 cell lines (Figure 2). 

### 2.7. Global Gene Expression Changes upon Treatment with Chemical Probes for KDM6B and PADI4

Treatment with the PADI4 inhibitor GSK484 led to 152 differentially expressed genes, of which 62 genes were downregulated and 90 were upregulated in A549 cancer cells, whereas the KDM6A/B inhibitor, GSK-J4, altered the expression of 190 genes, of which 56 genes were downregulated and 134 were upregulated in A549 cancer cells (FDR < 0.05 and LogFC > 1; <−1) (Appendix A). Functional enrichment analysis showed that treatment with PADI4 and KDM6A/B inhibitors was associated with processes linked to the collagen-containing extracellular matrix, extracellular matrix, extracellular space, cell periphery-related genes, and processes related to metastasis.

In cells treated with the PADI4 inhibitor, we found nine genes differentially regulated and six genes in cells treated with the KDM6A/B inhibitor, of which the following five genes were common among the treatments: the genes for Fibrinogen Alpha Chain (*FGA*), Nidogen 2 (*NID2*), Inter-Alpha-Trypsin Inhibitor Heavy Chain 2 (*ITIH2*), Peroxidasin (*PXDN*) and Heparin Binding EGF Like Growth Factor (*HBEGF*). All of the five genes common among the treatments are related to adhesion proteins, cell ligands, and protein stabilizing proteins of the extracellular matrix, suggesting that these genes participate in the regulation of metastasis (Figure 3). 

## 3. Discussion

Metastasis, one of the biggest problems of solid epithelial cancers, begins with the migration of tumor cells from the confined primary tumor to adjacent tissue, where tumor cells cross the basement membrane and lamina propria to invade the underlying connective tissue. Unlike normal epithelial cells, which undergo apoptosis when they lose contact with their native extracellular matrix, tumor cells develop mechanisms to detach from the primary tumor associated with epithelial organization, closely followed by the expression of mesenchymal markers [18]. These changes are the result of altered gene expression, which can be driven by epigenetic processes, thereby opening the possibility of affecting these changes by epigenetic regulation. Here, we performed a streamlined approach with in silico and in vitro analyses starting from 38 epigenetic targets to select the most relevant for lung cancer cell treatment and showed that the inhibition of PADI4 and KDM6B proteins controls the metastatic process, inhibiting cancer cell migration and invasion by altering their transcriptomes.

Protein-arginine deiminase Type-4 (PADI4) is a calcium-dependent enzyme that is known for its role in converting arginine to citrulline residues. Its downstream signaling has been studied in the progression of a variety of human cancers, but there is a lack of studies showing the efficacy of PADI4 in lung cancer [19,20]. Recently, Liu et al. (2019) demonstrated that PADI4 is overexpressed in lung cancer and contributes to cell growth and metastasis. Knockdown of PADI4 in A549 lung cancer cells resulted in a striking reduction in the EMT-associated Snail Family Transcriptional Repressor 1 (Snail1/mothers) against the decapentaplegic homolog ¾ transcriptional complex, which was consistent with alterations in migratory and invasive phenotypes of A549 lung cancer cells. On the other hand, the lysine demethylase 6B (KDM6B) is a histone demethylase that removes methyl groups from lysine and arginine residues on histone tails. It is a member of the Fe(II)- and α-ketoglutarate-dependent demethylases that activates gene expression by removing H3K27me3 marks on gene promoters [21]. KDM6B has been shown to be involved in tumor progression via the regulation of cell proliferation, migration, and senescence [22]. High levels of KDM6B induce the expression of mesenchymal genes, such as *Snail* and *Slug* (Snail Family Transcriptional Repressor 2), which promote *TGF-β*-induced (Transforming Growth Factor Beta 1) EMT and tumor metastasis [23]. Knockdown of KDM6B inhibited EMT induced by *TGF-β*, inhibiting breast cancer cell invasion [21]. Another study provided evidence of pulmonary metastasis of osteosarcoma in an in vivo model in which osteosarcoma cells were injected into the medullary cavity of nude mice. Intraperitoneal administration of GSK-J4 at concentrations above 5 mg/kg significantly inhibited the pulmonary metastasis of osteosarcoma cells in vivo. These results strongly suggest the potential of KDM6B as a target for highly metastatic osteosarcoma [24]. Thus, KDM6B may present a target for cancer metastasis. One point to consider is that GSK-J4 could also inhibit lysine demethylase 5B (KDM5B) histone demethylase and not only KDM6A/B. KDM5B has been implicated in several cancers, including NSCLC, and was recently described as a therapeutic target for cancer therapy [25]. However, GSK-J4 is more selectively potent for KDM6B than for KDM5B.

Interestingly, the treatment of cancer cells with non-cytotoxic doses of PADI4 and KDM6B inhibitors induced similar transcriptomic profiles, regulating genes related to cell adhesion and the extracellular matrix, which was associated with decreased capacity of cancer cells to invade and migrate in the in vitro model. For both inhibitors, upregulation of *FGA*, *NID2* and *ITIH2* genes, and downregulation of *PXDN* and *HBEGF*, was observed. Fibrinogen is an extracellular matrix protein composed of three polypeptide chains, fibrinogen alpha (FGA), beta (FGB), and gamma (FGG), and is involved in tumor angiogenesis and metastasis. *FGA* may play a suppressive role by inhibiting tumor growth and metastasis. FGA administration is considered a novel therapeutic approach to inhibit the growth and metastasis of lung adenocarcinoma [26]. Nidogen-2 (NID2) is ubiquitously present in the basement membrane and maintains its integrity and stability of the basement membrane by connecting laminin and collagen IV networks in the extracellular matrix (ECM). The restoration of *NID2* expression in cancer cells was shown to have a negative regulatory role in Epidermal Growth Factor Receptor (*EGFR*) and integrin signaling pathways, suggesting that *NID2* elicits in vitro migration/invasion suppression and in vivo metastasis inhibition effects through negative modulation of these two oncogenic pathways [27]. The other gene upregulated by both inhibitors was *ITIH2*, the inter-alpha-trypsin inhibitor 2, belonging to a family of plasma protease inhibitors, contributing to the stability of the extracellular matrix by covalently binding to hyaluronan. Loss or downregulation of *ITIH2* expression was observed in 70%, 71%, and 70% of breast, lung, and kidney tumors, respectively. In addition, careful densitometric evaluation of hybridization signals revealed downregulation in 56% of gastric cancers, 61% of rectal carcinomas, and 50% of prostate cancers [28].

Epigenetic inhibitors downregulated two genes in common: *PXDN* and *HB-EGF*. Peroxidasin (PXDN) is an extracellular matrix protein with peroxidase activity and has been reported to participate in epithelial mesenchymal transition processes, playing a promoting role in the proliferation, invasion, and migration of ovarian cancer cells through the regulation of *PI3K* (Phosphatidylinositol-4,5-Bisphosphate 3-Kinase Catalytic) pathway activation *Pl3k/Akt* (AKT Serine/Threonine Kinase), and is considered a potential target for therapy [29]. Heparin-bound epidermal growth factor-like growth factor (HB-EGF) is a member of the heparin-bound EGF family (Epidermal Growth Factor) and is more widely expressed in tumors than in normal tissues. HB-EGF can be produced in a membrane-anchored form (pro-HB-EGF) and further processed into a soluble form (s-HB-EGF), although a significant amount of pro-HB-EGF remains cleaved on the surface of the cell. In addition, wild-type s-HB-EGF or HB-EGF induced the expression and activity of the metalloproteases MMP-9 (Matrix Metallopeptidase 9) and MMP-3 (Matrix Metallopeptidase 3), leading to increased cell migration [30].

PADI4 inhibitor treatment in cancer cells downregulated four genes related to metastatic cancer phenotypes: Laminin Subunit Gamma 2 (LAMC2), C-X-C Motif Chemokine Ligand 8 (CXCL8), Niban Apoptosis Regulator 1 (FAM129A), and Pleckstrin 2 (PLEK2). Laminin Subunit Gamma 2 (LAMC2) is a subunit of the heterotrimeric glycoprotein laminin-332 (LAM-332, formerly laminin-5) consisting of α3, β3, and γ2 chains. Although LAMC2 is an important structural component of the epithelial basement membrane (BM) in various normal tissues, there is emerging evidence of a pathological role for the LAMC2 monomer in cancer [31]. *LAMC2* promotes migration and invasion via EMT, which is dependent on TGF-β1 and ZEB1 (Zinc Finger E-Box Binding Homeobox 1) integrin [31]. CXCL8, also known as interleukin-8 (IL-8), is a prototypic chemokine belonging to the CXC family and is responsible for the recruitment and activation of neutrophils and granulocytes to the site of inflammation [32]. Recent studies have shown that CXCL8 is essential for tumor cells to acquire and maintain this aggressive phenotype. A member of the family with sequence similarity 129, member A (FAM129A), inhibited apoptosis and promoted migration and proliferation in human cancers. One study revealed that *FAM129A* promoted tumor invasion and proliferation by upregulating the expression of MMP2 (Matrix Metallopeptidase 2) and cyclin D1, which was due to increased FAK (Protein Tyrosine Kinase 2) phosphorylation at Tyr 397 and Tyr 576. Overexpression of *FAM129A* was associated with tumor progression and predicted low survival of NSCLC patients [33]. Pleckstrin2 (PLEK2) is a 353 amino acid protein that is widely expressed in a variety of tissues and is highly expressed in NSCLC. *PLEK2* promotes NSCLC proliferation and metastasis via a BRD4-dependent *PI3K/AKT* signaling pathway that functions as an epigenetic reader and binds to acetylated lysine residues (KAc) that regulate chromatin structure and gene expression [34].

Treatment of cancer cells with a KDM6B inhibitor downregulated *FUT1* (Fucosyltransferase 1) gene, which is also related to metastasis. Fucosylation is a posttranslational modification that links fucose residues with protein- or lipid-linked oligosaccharides. Certain genes in the fucosylation pathway are aberrantly expressed in several types of cancer, including non-small cell lung cancer, and this aberrant expression is associated with poor prognosis in cancer patients. Fucosylation pathway genes, including fucosyltransferase 1/2/3/6/8 (*FUT1*, *FUT2*, *FUT3*, *FUT6*, *FUT8*) and GDP-L-fucose synthase (*TSTA3*), were correlated with poor patient survival in these patients. In this study, the inhibition of *FUT*s by 2F-peracetyl-fucose (2F-PAF) suppressed transforming growth *factor β* (*TGFβ*)-mediated *Smad3* (SMAD Family Member 3) phosphorylation and nuclear translocation in NSCLC cells. Furthermore, transwell wound healing and migration assays demonstrated that 2F-PAF inhibited the TGFβ-induced migration and invasion of NSCLC cells [35].

Our work emphasized the inhibition of important epigenetic targets related to the process of migration and invasion of tumor cells that favor cancer metastasis. Thus, these inhibitors have great potential to add to antitumor therapy, and can be added to other drugs already in clinical use, such as chemotherapy, immunotherapy and targeted therapy, contributing to the increase of antitumor effects, overcoming resistance to drugs already used and activation of the host’s immune response. Indeed, chemotherapy is still a traditional method in advanced cases, in which surgical excision is not possible, so the emergence of chemoresistance remains a major problem in cancer therapy. Thus, the combination of epigenetic drugs with other chemotherapeutics can not only promote a potent suppression of tumorigenesis, but also resensitize tumor cells to radiotherapy and chemotherapy [36]. Immunotherapy has been used as a promising candidate for both first- and second-line treatment in metastatic NSCLC. However, about 50% of NSCLC expressed PD-L1. There is no consensus predictive biomarker and resistance to immunotherapy can occur [37,38,39]. This fact limits the use of immunotherapy and overcoming immunotherapy resistance can be challenging due to the complex and dynamic interplay between malignant cells and the defense system. In the case of resistance, the epigenetic inhibitors could act as reactivating tumor suppressor genes and repress cancer cell growth. Some studies have shown that epigenetic inhibitors, such as BET, LSD1 and EZH2 inhibitors, are already used in combination with anti-PD1 therapy activating the antitumor immune response by increasing the persistence of T cells in the tumor microenvironment [3].

A study by Rohrbach and collaborators elucidates the relation between PAD4 activation and immune cells. PAD4 is expressed in granulocytes, which are essential for innate immunity and the formation of neutrophil extracellular traps (NETs). Anti-PAD4 therapies have been proposed for inflammatory and cancer conditions, but we need a better understanding regarding the role of neutrophils in cancer. The tumor microenvironment is composed of adaptive immune cells, which play important roles in tumor growth and metastasis [40]. Shi et al., 2020, transplanted Padi4 wild-type and Padi4-knocknout breast cancer cells into inguinal mammary fat pad areas of immunodeficient mice, which lacked functional T cells, B cells and NK, and found that tumor derived PADI4 facilitated metastasis, at least partially independent of the adaptative immune cells. Those findings together suggested that PADI4 inhibition can negatively affect the immune cells; however, the effects on metastatic cancer cells remained [41]. 

Lysine demethylase 6b (KDM6B) is essential for the generation and proper functioning of CD8+ effector T cells during acute infection and tumor eradication, being indispensable for proper effector functions and tumor protection, and KDM6B inhibition exhibits a memory-defective T cell response. Therefore, KDM6B may act as an epigenetic modulator of CD8+ T cell fate determination by regulating effector-associated gene expression and chromatin accessibility [42]. As members of the KDM6 family have been therapeutic targets for several cancers, it is necessary to properly understand their intrinsic role in T cell function. More studies are necessary to better understand the interaction between epigenetic protein inhibition and immunotherapy.

## 4. Materials and Methods

A streamlined set of in silico analyses coupled with in vitro analyses (Figure 4) was performed to evaluate and rank potential epigenetic targets based on epigenetic probes from the SGC (https://www.thesgc.org, accessed on 25 January 2019). The screening was based on a list of available epigenetic inhibitors from SGC, followed by an analysis of the association of the epigenetic target with NSCLC survival, subsequent selection by a systematic review of potential new cancer targets, in silico analysis of protein expression in NSCLC cell lines, and real-time qPCR expression to evaluate target expression in the cell lines.

### 4.1. In Silico Analysis for Determination of Potential Epigenetic Targets in Patients with Non-Small Cell Lung Cancer

The Kaplan-Meier Plotter software [43] (http://kmplot.com/analysis, accessed on 4 February 2019).) was powered with public data from 14 repositories with information on gene expression and clinical samples, totaling 2438 cases of NSCLC. Thirty-eight epigenetic targets (Appendix A) were individually analyzed for their association with the survival rates of 590 patients diagnosed with adenocarcinoma and 492 patients with squamous cell carcinoma (Table 2). The patient selection criteria considered the histological types, grouping them into adenocarcinoma and squamous cell carcinoma, patients in stages I, II, III, and IV of the disease, of both sexes, smokers and non-smokers, and if patients had started any type of treatment, such as surgery, chemotherapy, and radiotherapy. The follow-up time for each patient was evaluated from the time of diagnosis to the time of death. For all analyses, the results (*p* < 0.05) were considered, according to the log-rank test (chi-square), to compare whether there was a statistical difference between the curves of high and low gene expression and the use of the hazard ratio (HR) with a 95% confidence interval. A hazard ratio equal to one means no association between treatments, a rate greater than one suggests an increase in risk and below one suggests a decrease in risk.

For the selection of the cases, the following inclusion criteria were applied: use of cases that presented patient survival; quality control of array chips excluding chips with outliers (>95% of total arrays) from analysis for any of the following parameters: percentage of calls present, background, rawQ, bioB-/C-/D-spikes, *GAPDH* (Glyceraldehyde-3-Phosphate Dehydrogenase) and *ACTB* (Actin Beta) 3 ratio for 5. As recommended by the authors, the Jetset Best probe set was always used to analyze the expression of genes of interest, and a high and low expression group based on quartiles (25%, Q1 × Q4) was created for survival analysis. Analyses were performed independently for adenocarcinoma and squamous cell carcinoma cases. Statistical analysis was performed using univariate Cox regression, generating *p*-values and hazard ratios.

### 4.2. Systematic Literature Review for Selection of New Epigenetic Targets for NSCLC

An independent systematic literature review was performed for each epigenetic target in the PubMed database (https://www.ncbi.nlm.nih.gov/pubmed, accessed on 7 February 2019). For literature selection, a specific set of keywords was used, presenting the abbreviation of the name of the epigenetic target and the term “lung cancer,” as being mandatory in the titles in order to find studies that specifically evaluated epigenetic targets and lung cancer. The analysis was performed on publications published from 2000 to 2019.

### 4.3. In Silico Analysis of Epigenetic Target Expression

The CellExpress software [44] (http://cellexpress.cgm.ntu.edu.tw, accessed on 11 March 2019) was used to perform gene expression analysis on more than 4000 tumor cell lines and clinical samples obtained from public datasets. For expression analysis, the databases of gene expression studies of cell lines NCI-60 Human Tumor Cell Lines Screen (GSE32474), Cancer Cell Line Encyclopedia-CCLE (GSE36133), and Sanger Cell Line Project (GSE68950) were used. Microarray data obtained on the same platform were normalized using a quartile normalization algorithm to remove systematic biases. Expression data from the GSE36133 study from the CCLE database were used, which presented a more complete list of cell lines of interest, using the selection of the Jetset Best probe and the expression of the four potential epigenetic targets in lung adenocarcinoma cell lines. The probes selected to assess gene expression levels in cells were from the Jetset Best probe set, being (41386_at) for *KDM6B* (212512_at) for *CARM1*, (201353_at) for *BAZ2A* and (220001_at) for *PADI4*, which were also used in the Kaplan-Meier survival analysis. The endogenous gene, *GAPDH,* was used to normalize the expression levels of the genes of interest. The results were generated by calculating the relative expression, which showed similar gene expression values between the analyzed cells. Evaluation of the expression of targets in healthy lung tissue was performed using the CellNavigator software (https://medicalgenomics.org/rna_seq_atlas, accessed on 15 March 2019) through microarray analysis coupled to the RNA-seq Atlas platform through the Human Genome Set U133 (HG-U133). Background correction, normalization and summarization was performed by applying the frma function from the fRMA package to AffyBatch with default options. The Z-Score transformation was calculated using the barcode function of the fRMA package to standardize gene values from the Microarray data. The barcode options were set for the corresponding platform and the output method was set to ‘z-score’. Then, the Z-Scores were averaged for each tissue and each pathological state (healthy, cancer), Z-Score > 5 suggests that the gene is expressed in that tissue. Finally, the Z-Score was averaged for each tissue and state (healthy, cancer) and stored in the PostgreSQL database.

### 4.4. Cell Culture

The lung cancer cell lines A549, H1568, and H2126 were donated by Dr. Lucy M. Anderson from the Laboratory of Comparative Carcinogenesis at the Frederick National Laboratory for Cancer Research, Frederick, MD, USA, and maintained as previously described [45]. Briefly, cell lines were maintained in 75 cm^2^ flasks at 37 °C and 5% CO_2_ in RPMI-1640 medium supplemented with 10% fetal bovine serum (FBS) and 1% antibiotic/antimycotic (Pen-Strep). Cell passaging was performed when the cells were 85% confluent using TrypLE^TM^ Express trypsin. Culture evolution was evaluated daily using optical microscopy (Axio Vert A1, Zeiss, Jena, Germany). All the reagents used for cell culture were purchased from Thermo Fisher Scientific (USA). All cell lines were authenticated at the Laboratory of Molecular Diagnosis of the Cancer Hospital of Barretos (Hospital de Amor HA) as previously reported [46] before the experiments and were free for Mycoplasma spp. by real-time PCR (Myco-Sniff-Valid™ Mycoplasma PCR Detection Kit).

### 4.5. RNA Isolation and Reverse Transcription-Quantitative Polymerase Chain Reaction (RT-qPCR)

Total RNA was extracted from A549, H1568, and H2126 cell lines using TRIzol (Invitrogen; Thermo Fisher Scientific, Waltham, MA, USA) according to the manufacturer’s protocol. cDNA was synthesized from total RNA (1000 ng) using a High-Capacity cDNA Reverse Transcription kit (Applied Biosystems; Thermo Fisher Scientific, Inc.) using the following parameters: 25 °C for 10 min, 37 °C for 120 min, and 85 °C for 5 min. qPCR was performed using the SYBR Master Mix (Roche Diagnostics, Basel, Switzerland). Gene expression analyses were performed by real-time PCR using the StepOne System (Thermo Fisher Scientific). Specific primers were designed using Primer-BLAST [47] and dimers and hairpins were verified using AutoDimer software [48]. Primers were also analyzed using in silico polymerase chain reaction (PCR) (https://genome.ucsc.edu/cgi-bin/hgPcr, accessed on 7 March 2019) to confirm specificity. The primer sequences are listed in (Appendix A). PCR was carried out using Fast SYBR Green Master Mix in a final volume of 10 µL. The conditions for quantitative PCR were as follows: 95 °C for 20 s; 40 cycles at 95 °C for 3 s for denaturation and 60 °C for 30 s for anneal/extension; melt curve analysis was performed at 95 °C for 15 s and 60 °C for 60 s. The housekeeping gene used was *18 s* ribosomal RNA, and the analysis of relative gene expression data was performed according to the ΔΔCt method [49]. The experiments were performed twice and in triplicate. All reagents were purchased from Thermo Fisher Scientific. The experiment was performed twice and in triplicate.

### 4.6. Epigenetic Probes Cell Viability Assay

The epigenetic probes (Cayman Chemical, Ann Arbor, MI, USA) were dissolved in dimethylsulfoxide (DMSO) to a concentration of 20 mM. A549 cells were seeded at 5000/well in 96-well plates (Corning, NY, USA) containing 100 µL of supplemented media, as described previously. After 24 h, the medium was replaced with fresh culture medium containing different concentrations of epigenetic probes, ranging from 10 µM to 13.72 nM. Epigenetic probes were added in six replicates per concentration and the experiments were performed in triplicate. After 72 h, 10 µL of 3-(4.5-dimethylthiazol-2-yl)-2.5-diphenyl tetrazolium bromide (MTT, 5 mg/mL) was added to each well and formazan crystals were produced over a 2 h incubation period. One hundred microliters of DMSO were added to dissolve the crystals. The optical density at 540 nm was measured using Fluorstar Optima (BMG Labtech, Ortenberg, Germany). The concentration of the compound corresponding to the IC_50_ was calculated using a nonlinear regression test performed in GraphPad Prism (version 6.00 for Windows, GraphPad Software, USA).

### 4.7. Invasion Assay

The cells were treated with DMSO (control) or 100 nM of TP-064, GSK2801, GSK-J4, and GSK484 for 72 h. Cells were cultured for 24 h in serum-free medium. Transwell inserts were placed in 24 well plates and filled with 100 μL of ECM gel (Sigma Aldrich Saint Louis, MO, USA) in RPMI-40 medium (1:5). After, 2 × 10^4^; A549 and 2.5 × 10^4^ H1568 cells were resuspended in 100 μL serum-free medium and plated on inserts. The bottom well was filled with 600 μL of RPMI-40 medium supplemented with 20% fetal bovine serum (FBS), used as a chemoattractant, and after 48 h, a cotton swab was used to remove non-invasive cells from the top of the inserts. As a fixative, 5% glutaraldehyde was used for 10 min at room temperature and inserts were stained with 1% crystal violet in 2% ethanol for 20 min. The invasive cells were observed and photographed under an optical microscope in five random fields at 100× magnification using the ZEISS ZEN 2 Microscope Software (ZEISS, Germany). Finally, the invasive cells were counted using ImageJ software version 1.8.0_112 [46]. The experiment was performed thrice in duplicate.

### 4.8. RNAseq Data Generation

To assess the genes affected by treatment with GSK 484 (PADI4) and GSK-J4 (KDM6A/B), A549 cells were treated with 100 nM of these inhibitors. Duplicates of each treatment and control group were prepared. A549 cells were treated with 100 nM GSK-J4, GSK484, and DMSO (control) for 72 h and then RNA was extracted using TRIzol (Invitrogen; Thermo Fisher Scientific, Inc.). RNA quality and quantity were assessed using automated capillary gel electrophoresis on a Bioanalyzer 2100 with RNA 6000 Nano Labchips, according to the manufacturer’s instructions (Agilent Technologies, Cork, Ireland). Only samples that presented an RNA integrity number (RIN) higher than 8.0 were considered for sequencing. RNA libraries were constructed using the TruSeq™ Stranded mRNA LT Sample Prep Protocol and sequenced on an Illumina HiSeq platform. 2500 equipment in HiSeq Flow Cell v4 using a HiSeq SBS Kit v4 (2 × 100 bp).

### 4.9. Alignment and Differential Expression

Sequencing quality was evaluated using FastQC software (http://www.bioinformatics.babraham.ac.uk/projects/fastqc, accessed on 3 March 2019), and no additional filtering was performed. Sequence alignment against the human reference genome (GRHC38) was performed using STAR [50], according to standard parameters and including the annotation file (Ensembl release 89). Secondary alignments, duplicated reads, and reads failing vendor quality checks were removed using Samtools [51]. The alignment quality was confirmed using Qualimap [52]. Gene expression was estimated by read counts using HTseq [53] and normalized to counts per million reads (CPM). Only genes presenting at least one CPM in at least four samples were retained for differential expression (DE) analysis. DE was performed using the EdgeR package [54] in the R environment based on a negative binomial distribution. The Benjamini-Hochberg procedure was used to control the false discovery rate (FDR), and transcripts with FDR ≤ 0.05, and log-fold change (LogFC) > 1; <−1 were considered differentially expressed (DE). Functional enrichment analysis of the DE genes was performed using STRING [55,56].

### 4.10. Statistical Analysis

The IC_50_ was calculated using a nonlinear regression test. Gene expression was analyzed by one-way ANOVA with Tukey’s post-hoc test. One-way ANOVA followed by Student’s *t*-test was used for invasion assays. For functional enrichment analyses, *p*-values were adjusted for multiple tests, and the Benjamin and Hochberg method was used to test multiple categories in a group of functional gene sets. Differences were considered statistically significant at *p * < 0.05.

## 5. Conclusions

In summary, a streamlined approach of in silico and in vitro experiments allowed us to select, from 38 different epigenetic targets, the two most promising candidates for NSCLC drug development: PADI4 (GSK 484) and KDM6B (GSK-J4). The inhibition of these epigenetic proteins regulates molecular pathways in NSCLC, affecting the ability of cancer cells to migrate and invade, thereby controlling the metastatic cascade. Treatment with the identified inhibitors regulates common genes linked to tumor metastasis. 

## Figures and Tables

**Figure 1 ijms-23-11911-f001:**
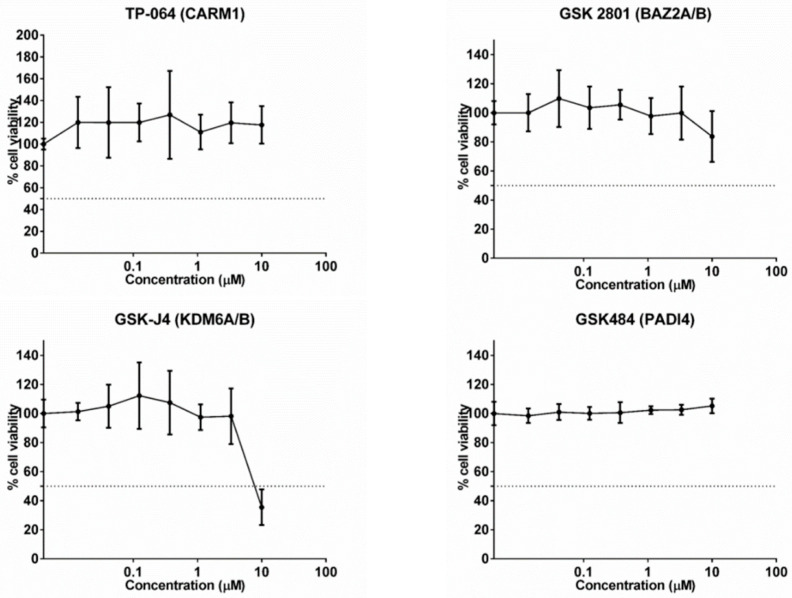
Cytotoxic potential of the 4 specific epigenetic molecules in the A549 cell line. The GSK-J4 (KDM6A/B) molecule had an IC_50_ value of 8.21 µM, the others had a value greater than 10 µm.

**Figure 2 ijms-23-11911-f002:**
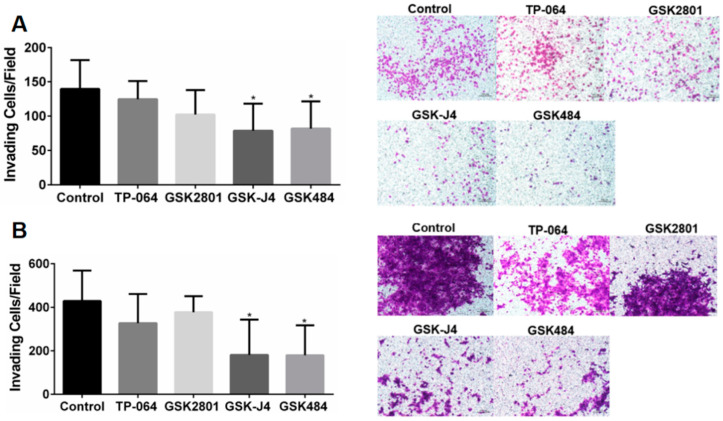
(**A**) Analysis of cell invasiveness after treatment with epigenetic molecules in the A549 cell line. (**B**) Analysis of cell invasiveness after treatment with epigenetic molecules in the H1568 cell line. Transwell invasion test evaluated after 48 h (* *p* < 0.05, One-way ANOVA followed by Student’s *t*-tests).

**Figure 3 ijms-23-11911-f003:**
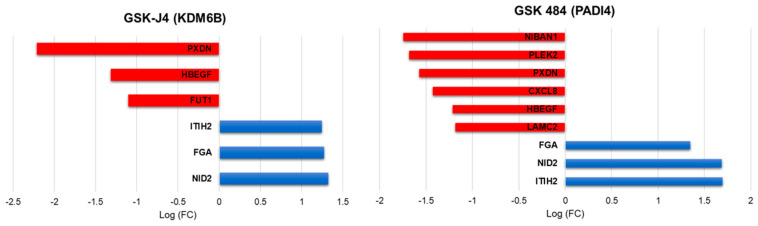
Representation of log-fold change value of EMT-related gene regulation after treatment with PADI4 and KDM6B inhibitors.

**Figure 4 ijms-23-11911-f004:**
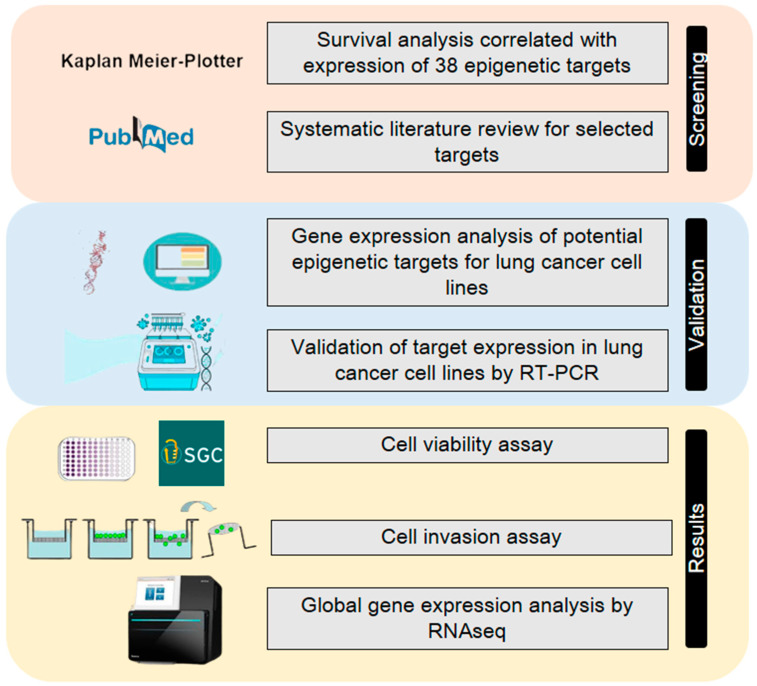
The streamlined approach using in silico and in vitro experiments. The initial screening was performed using publicly available data of lung cancer samples from TCGA using the KMplotter. Then, a systematic review of the significant candidates was performed to exclude the already described targets in the literature for lung cancer. The potential candidates were evaluated by gene expression in cancer cell lines, first, in silico and, then, experimentally in cancer cells. Lastly, experiments for cytotoxicity, an invasion assay and transcriptomic analyses were performed.

**Table 1 ijms-23-11911-t001:** The eight significant potential epigenetic targets selected by inverse association with survival rates of patients diagnosed with pulmonary adenocarcinoma. The survival medians of the low and high expression groups of the targets were also analyzed.

Epigenetic Targets	Enzyme Class	*p* Value	Hazard Ratio (HR)	Low Expression Survival (Median in Months)	High Expression Survival (Median in Months)
PRMT1	Methyltransferase	5.8 × 10^−9^	(HR = 3.22; IC = 95%; 2.12–4.88)	75	21
KDM6B	Demethylase	6.3 × 10^−9^	(HR = 2.81; IC = 95%; 1.95–4.03)	150	34
CARM1	Methyltransferase	9.2 × 10^−8^	(HR = 2.73; IC = 95%; 1.86–4.00)	48	18
BAZ2A	Bromodomain	7.8 × 10^−6^	(HR = 2.23; IC = 95%; 1.56–3.20)	175	52
BRD4	Bromodomain	0.0025	(HR = 1.81; IC = 95%; 1.23–2.68)	117	69
EZH2	Methyltransferase	0.024	(HR = 1.50; IC = 95%; 1.05–2.13)	126	70
PADI4	Deiminase	0.025	(HR = 1.47; IC = 95%; 1.05–2.06)	107	80
BRD9	Bromodomain	0.033	(HR = 1. 45; IC = 95%; 1.03–2.05)	103	52

**Table 2 ijms-23-11911-t002:** Clinical data from 590 patients with adenocarcinoma evaluated for expression of epigenetic targets with the Kaplan-Meier Plotter.

Clinical Data	*n*	Freq. (%)
Histology		
Adenocarcinoma	590	−100%
Stage		
1	277	(46.95%)
2	115	(19.49 %)
3	16	(2.71 %)
4	4	(0.68%)
Staging		
T1	123	(20.85%)
T2	103	(17.46%)
T3	4	(0.68%)
T4	0	0%
N0	184	(31.19%)
N1	42	(7.12%)
N2	3	(0.51%)
M0	229	(38.81%)
M1	1	(0.17%)
Gender		
Women	247	(41.86%)
Men	289	(48.98%)
Smoking history		
Exclude those never smoked	180	(30.51%)
Only those never smoked	92	(15.59%)
Surgery success		
Only surgical margins negative	127	(21.53%)
Chemotherapy		
Yes	14	(2.37%)
No	8	(1.36%)

## Data Availability

Not applicable.

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
