# Peer review of "A Screening of Epigenetic Therapeutic Targets for Non-Small Cell Lung Cancer Reveals PADI4 and KDM6B as Promising Candidates"

_ijms, 2022, doi:10.3390/ijms231911911_

Round 1

Reviewer 1 Report

The manuscript titled “A screening of epigenetic therapeutic targets for non-small cell lung cancer reveals PADI4 and KDM6B as promising candidates” described PADI4 and KAM6B as potential epigenetic targets for treating lung adenocarcinoma through correlation of gene expression and survival of patients. Lung cancer is still the leading cause of cancer-related death with ONLY 21% of 5-year survival rate due to the large proportion (57%) diagnosed at metastatic stage. The median survival for even high expression of PADI4 was 80 months! Besides PADI4, high expression of BRD4 and EZH2 were also reported to have more than 5-year median survival. The findings were not consistent with current understanding. The treatment background, gene mutation status, age, smoking status, etc. for the patients may need report. In addition, the median survival (107 months) for low expression of PADI4 was not obviously changed, making PADI4 maybe not an interesting target. Furthermore, the in vitro experiments did not support the descriptive finding and there was no logic how the experiments were designed. In general, the story was poorly designed and written.

More comments:

1.      The background for the patients needs to be described in detail in a table.

2.      On page 3 in row 91, it was not clear why four potential epigenetic targets were selected, since PADI4 and KAM6B was not reported.

3.      In 2.4, the high expression should be relative to normal bronchial cell line or tissue.

4.      As described in 2.3, H2126 cell line was collected from metastatic sites, assuming the cell line was highly metastatic. However, in 2.6, the author did not detect the invasion and metastasis with the cell line due to previous conflict work.

5.       In Figure 1, only A549 cell line was assessed.

6.      As shown in Figure 1, only KDM6A/B inhibitor but not PADI4 inhibitor showed limited cytotoxicity to A549 cell line, it was odd to compare the various gene expression changes in A549 by treatment with these two inhibitors. The findings upon them was not convincing.

Author Response

Dear Reviewer/Editor,

We prepare a file with all the points raised by both reviewers. The document is attached.  

Reviewer 2 Report

I thank the authors for allowing me to review this article, having an original approach of prognostic factors in lung cancer under the angle of epigenetics, important target in cancer.

They demonstrated an impact on gene expression, cell invasion and cytostatic potential of both inhibitors. But what would be the strategy with chemotherapy, immunotherapy and targeted therapies?

Moreover, if this molecule inhibits cell invasion at the level of lung cancer cells, what about the immune cells, T cells, important today in the management of lung cancer?

Author Response

(The authors gave the same response as above.)

Round 2

Reviewer 1 Report

No further suggestions.